# Multiscale Modelling and Analysis for Design and Development of a High-Precision Aerostatic Bearing Slideway and Its Digital Twin

**Ning Gou** [1],*[ID]**, Kai Cheng** [1] **and Dehong Huo** [2][ID]

1 College of Engineering, Design and Physical Sciences, Brunel University London, Uxbridge UB8 3PH, UK; Kai.Cheng@brunel.ac.uk
2 School of Engineering, Newcastle University, Newcastle NE1 7RU, UK; dehong.huo@newcastle.ac.uk
* Correspondence: Ning.Gou@brunel.ac.uk

**Abstract:** Aerostatic bearing slideways have been increasingly applied in the precision engineering industry and other high-tech sectors over the last two decades or so, due to their considerable advantages over mechanical slideways in terms of high motion accuracy, high speeds, low friction, and environment-friendly operations. However, new challenges in air bearings design and analysis have been occurring and often imposed along the journeys. An industrial-feasible approach for the design and development of aerostatic bearing slideways as standard engineering products is essential and much needed particularly for addressing their rapid demands in diverse precision engineering sectors, and better applications and services in a continuous sustainable manner. This paper presents the multiscale modelling and analysis-based approach for design and development of the aerostatic bearing slideways and its digital twin. The multiscale modelling and analysis and the associated simulation development can be the kernel of the digital twin, which cover the mechanical design, direct drive and control, dynamics tuning of the slideway, and their entire mechatronic system integration. Using this approach and implementation, the performance of an aerostatic bearing slideway can be predicted and assessed in the process. The implementation perspectives for the sideway digital twin are presented and discussed in steps. The digital simulations and digital twin system can be fundamentally important for continuously improving the design and development of aerostatic bearing slideways, and their applications and services in the context of industry 4.0 and beyond.

**Keywords:** multiscale modelling; aerostatic bearing slideway; air bearing design; digital twin; Computational Fluid Dynamics (CFD) simulation; direct drive



## 1. Introduction

Engineering modelling and simulation have been widely applied over the past three decades or so [1]. For instance, macroscopic models are used to reveal the relationship between geometrical structure and kinematic, dynamic or mechanical performance in machine design. While the influences of surface features of machine components are often discussed through mesoscopic modelling, meanwhile, much R&D attention has been paid to the crystal morphology of the component materials in microscopic modelling or the molecular dynamics. For precision and ultraprecision machines, however, the factors and features from the machine itself, the component metalogical surface or tolerance, and the machine assembly clearance in multiple scales will have effects not only on themselves, but also collectively on the performance of the entire machine. Therefore, as the machine performance and precision are ever increasing, it is essentially important to develop a multiscale modelling and analysis-based approach for design and development of high precision machines, particularly for overcoming the limitations in design and building of the machine systems for generating nanometric surfaces on an industrial scale. Furthermore, such an approach combining with digital twin will likely lead to continuous

improvement of the performance of the machine system in operations and services, as well as their design and development in a predictable, producible, and highly productive manner in the competitive engineering marketplace [2].

There has been a considerable amount of research efforts on multiscale modelling applied to various engineering fields over the last two decades or earlier. Weinan [3] summarized several classic multiscale modelling methods and the related principles from applied mathematics aspects, and Tadmor, et al. developed the classic nonlocal quasi-continuum method (QC) for simulating isolated defects including dislocations and cracks in single crystals [4]. Over the years in parallel, the multiscale modelling and analysis and the associated simulations development have been widely attempted in applications for material science [5], Biomedical Engineering [6], and Mechanics [7], etc.

However, lots of research and development related to multiscale modelling and simulations often comes down to the individual application case driven, i.e., focused on one specific case. According to the statistics from Clarivate Analytics [8], there are few researches published under the keywords of 'multiscale modelling' and 'air bearing' or 'aerostatic slideway'. In precision engineering, aerostatic bearings have emerged as a key enabling technology particularly for high-precision machines and facilities [9,10], while their design and analysis have gone through the methodology journey from experimental-based analysis, using differentiation equations method as the design aid, FEA method, CFD method, and/or FEA/CFD simulations combined. However, as precision products, the air-bearings slideways or spindles are becoming technologically more complex because of the high-precision mechanical components, electrical direct drive, encoder devices as positioning feedback, and advanced control algorithms involved. This further imposes challenges on the design and development of these products particularly towards even higher precision and application performance requirements. The multiscale modelling and analysis, and the associated advanced simulations and digital twin integration, have substantial potentials and impacts on aerostatic bearing slideways design and development by likely helping in-process or remote tuning, tracking error prediction, higher precision enhancement, and continuous improvement of the product, etc. [11].

In this paper, the development of multiscale modelling and simulation for the aerostatic bearing slideway is presented firstly while focused on the key influence factors working at different scales in particular, e.g., the effect of air film thickness in the microscale, orifices' pattern and distribution in mesoscale, the slideway structure in macroscale, and the slideway's application performance as a whole. Then simulation development with COMSOL Multiphysics is described to analyze the static performance of the slideway. Furthermore, through the simulations on a linear motor control system for the aerostatic bearing slideway with MATLAB/SIMULINK integrated with COMSOL, the dynamic performance of the slideway designed is explored and further analyzed. These simulations are used as the kernel for further developing the digital twin system of the slideway. Finally, the paper is concluded with further discussion on implementation perspectives of the simulation—digital twin system, and their application potentials.

## 2. Design of the Aerostatic Bearing Slideway and Its Digital Twin

As the key component of ultra-precision machine tools, aerostatic bearing slideway relies on an external source of high-pressure air to generate appropriate load-carrying capacity. The principle of aerostatic bearing slideway is to generate an air film between the moving part, which is derived by a linear motor and the stationary part; this air film acts as the lubricant during the relative motion, as shown in Figure 1. There are numerous orifices in every direction on the slide surface; gas outputted by an air pump passes through these orifices and can form a stable and rigid air film. The main advantages in lubricating a slideway with compressed air are the low friction, which is almost zero at low speed, although the stiffness of air film is normally lower compared with other lubrication material like oil, but it generates lower heat and surface adhesions at the same time, which give the slideway an extremely long service life and low ratio damping.

The study result of Wardle [9] showed that aerostatic bearing slideway can produce the lowest motion errors of any slideway type in industrial applications, thus enabling air bearing slideways to compete very favorably with other slideway types in ultra-precision machining applications, where high dimensional precision and surface quality are required.

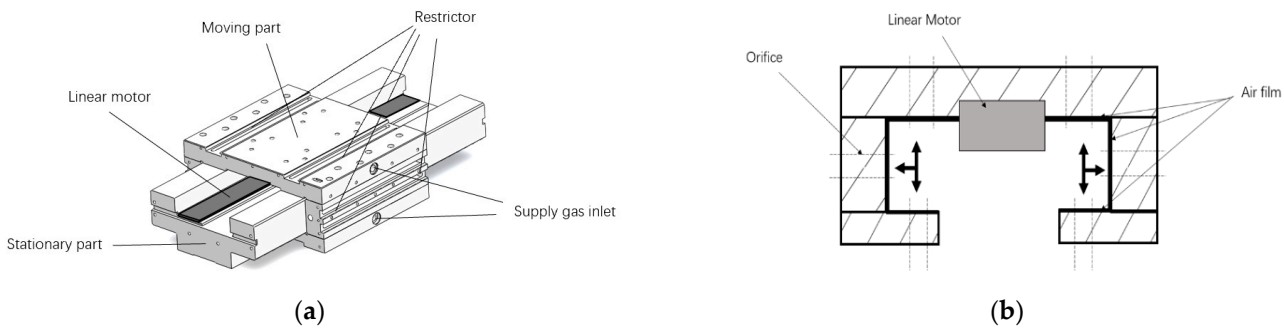

(**a**)                   (**b**)

**Figure 1.** The aerostatic bearing slideway designed: (**a**) key components; (**b**) Schematic illustration.

In application scenarios, the static performance of aerostatic bearing slide is critically important. As the density of air is very low, the lubricating air film thickness or internal clearance within an aerostatic bearing slideway must be kept in micron level so that a reasonable working pressure can be maintained within the slideway and the air consumption can be kept down to an acceptable level due to economic consideration. The recognized reasonable air film thicknesses are in the 5–20 μm range [10]; with such small clearances, every small deviation or change in parameter will have great influence on the performance of the slideway.

Figure 1 also illustrates the aerostatic bearing slideway designed and developed by the authors' team. The parameter of each part, the layout, amount, diameter and pattern of the orifices, the supply pressure of air, the thickness of the air film etc. will affect the static stiffness and load capacity of the slideway on various scales; details of these influence factors will be discussed in subsequent sections.

In order to achieve and maintain extremely high accuracy and stability in different working environments, the control system of aerostatic bearing slideway products often needs to be re-debugged before and after work; thus, in order to provide an appropriate solution, this study aims to the discuss the integrated approach combining multiscale modelling-based simulation and digital twin technology in the development and design of an aerostatic bearing slideway. As an important part of the German "Industry 4.0" national strategic initiative, digital twin technology uses a physical model, sensor update, operation history, and other data to integrate a multi-disciplinary, multi-physical, multi-scale simulation process to complete the "twin" in virtual space, so as to reflect the status of corresponding "physical twin" in real time. Currently, it is widely used in product design, medical analysis, engineering construction, and other fields.

In this study, the grating ruler is used as an in-process sensor to realize the communication between digital and physical slideway, it transports feedback signals of the dynamic characteristics such as the position and velocity of the moving part in real time, so as to realize the remote monitoring, adjustment, and even error prevention of the aerostatic bearing slideway.

In the very first development stage, the digital twin consists only of an exact CAD representation of the aerostatic bearing slideway. The air film is regarded as rigid with solid size. Next stages include the combination of finite element method and state-space method in order to accurately simulate the entire aerostatic bearing slideway's static and dynamic performance, and the function of remote monitoring and tuning are realized with the help of an in-process sensor. In following sections, details of the simulation work, which are developed in the collaboration with Solidworks, COMSOL Multiphysics and

MATLAB-based CFD module, fluid-structure interaction module, SIMULINK module etc. will be discussed separately.

## 3. Multiscale Modelling Applied to the Aerostatic Bearing Design and Analysis

In order to build the appropriate multiscale modellings, it is necessary to discuss the influence factors in different scales. The equations and theories have been divided into different length scales against the computational time as illustrated in Figure 2 [3]. Those fundamental theories and the associated implementation approaches are used to solve industrial problems on various orders of magnitude. The following description attempts to discuss the influence factor to the slideway's performance of each scale.

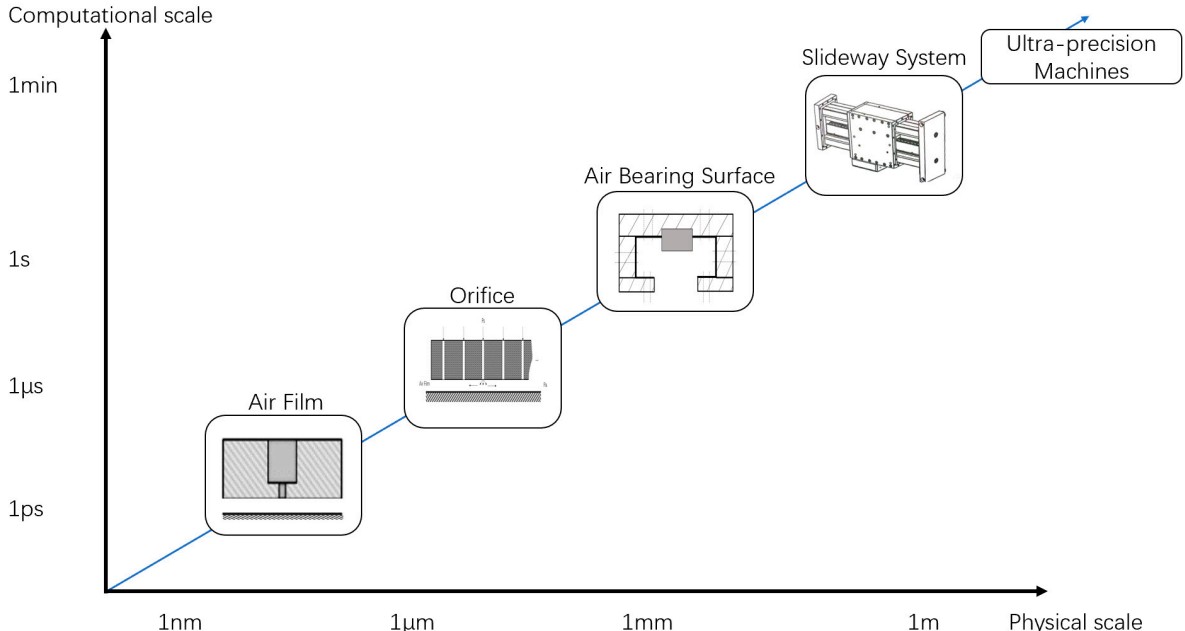

**Figure 2.** Different scales against the computational time.

The performance of aerostatic bearing slideway can be roughly divided into two types: static performance and dynamic performance. The static performance refers to the mechanical characteristics of the slideway in working environment, including static load capacity, static stiffness, and the volume of gas flow. The dynamic performance refers to the non-steady state of the slideway, including motion accuracy, stability, response speed etc. Since these two kinds of performance are reflected in different physical systems, they will be discussed separately.

### 3.1. Modelling at the Macroscale

The design and simulation in macroscale focus on the fluid–structure interaction in the lubrication domain; in this study, gas is defined as incompressible fluid, so it follows the law of conservation of mass, conservation of momentum, liquid continuity, and continuum mechanics etc. The load capacity of a slideway can be obtained by integrating the pressure of the entire air film surface. The related calculation formula is:

$$W_0 = \int_A (p - p_a) dA \tag{1}$$

where:
$W_0$—the load capacity of a gas film surface;
$A$—the surface area of air film;
$p$—the pressure distribution; and

$p_a$—the atmospheric pressure.

The static stiffness of a slideway illustrates the extent of change in the load capacity when the air film thickness changes under an applied load, which can be calculated by:

$$K_0 = \frac{W_1 - W_2}{\Delta h} \tag{2}$$

where:
$K_0$—the static stiffness of aerostatic bearing slideway;
$W_1$, $W_2$—the load capacity with different air film thickness; and
$\Delta h$—the change of film thickness.

Meanwhile, the air flow volume can be calculated by:

$$Q = \int_A \rho v \, dA \tag{3}$$

where:
$Q$—the air flow volume;
$\rho$—the density of air;
$v$—the velocity of air; and
$A$—the area of export border.

### 3.2. Modelling at the Mesoscale

Several studies have discussed the influence of restrictor on the load capacity and stiffness of an aerostatic bearing slideway in mesoscale; for example, Gao, et al. showed the relationship between the type of restrictor and the air film performance under a certain supply pressure [12], as shown in Figure 3. In this study, due to the consideration of technical, economical and practical consideration, the industrial collaboration partner implemented annular orifice as the initial choice; the influence factors like diameter, pattern structure, and amount of the selected orifice type are discussed separately in the following sections.

| Restrictor Type | Load Capacity | Stiffness | Stability | Gas Consumption | Manufacture |
|---|---|---|---|---|---|
| Annular orifice | Low | Low | Fair | Small | Easy |
| Simple orifice | High | High | Poor | Small | Easy |
| Slot | Medium | Medium | Good | Large | Medium |
| Groove | High | High | Good | Medium | Hard |
| Porous | High | High | Excellent | Large | Hard |

**Figure 3.** Characteristics of different types of restrictor types. Reprinted with permission from ref. [12]. Copyright 2019 Elsevier.

### 3.3. Modelling at the Microscale

In micrometre–millimetre scale, the static performance of aerostatic bearing slideway is mainly influenced by the parameter of air film. The operating principle of orifice aerostatic bearing slideway is shown in Figure 4, the air film thickness $h_0$ is normally about eight times smaller compared with the dimension of human hair, nevertheless, it still plays an important role in improving the load capacity and motion stability of the entire machine tool.

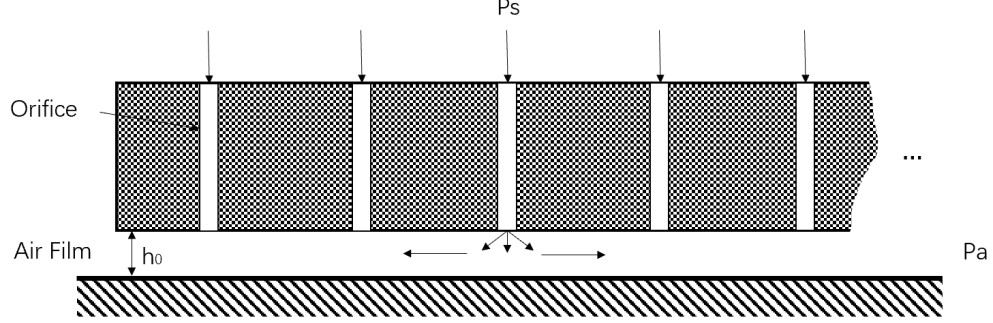

**Figure 4.** Operating principle of annular orifice aerostatic bearing slideway.

As a classic algorithm that describes the pressure distribution of compressible viscous fluid, the Navier–Stokes equation is here introduced:

$$
\begin{aligned}
\rho \frac{du}{dt} &= \rho f_x - \frac{\partial P}{\partial x} + \frac{\partial}{\partial x}\left\{\mu\left[2\frac{\partial u}{\partial x} - \frac{2}{3}\left(\frac{\partial u}{\partial x} + \frac{\partial v}{\partial y} + \frac{\partial w}{\partial z}\right)\right]\right\} + \frac{\partial}{\partial y}\left[\mu\left(\frac{\partial u}{\partial y} + \frac{\partial v}{\partial x}\right)\right] + \frac{\partial}{\partial z}\left[\mu\left(\frac{\partial u}{\partial z} + \frac{\partial w}{\partial x}\right)\right] \\
\rho \frac{dv}{dt} &= \rho f_y - \frac{\partial P}{\partial y} + \frac{\partial}{\partial y}\left\{\mu\left[2\frac{\partial u}{\partial y} - \frac{2}{3}\left(\frac{\partial u}{\partial x} + \frac{\partial v}{\partial y} + \frac{\partial w}{\partial z}\right)\right]\right\} + \frac{\partial}{\partial z}\left[\mu\left(\frac{\partial v}{\partial z} + \frac{\partial w}{\partial y}\right)\right] + \frac{\partial}{\partial x}\left[\mu\left(\frac{\partial u}{\partial y} + \frac{\partial v}{\partial x}\right)\right] \\
\rho \frac{dw}{dt} &= \rho f_z - \frac{\partial P}{\partial z} + \frac{\partial}{\partial z}\left\{\mu\left[2\frac{\partial u}{\partial z} - \frac{2}{3}\left(\frac{\partial u}{\partial x} + \frac{\partial v}{\partial y} + \frac{\partial w}{\partial z}\right)\right]\right\} + \frac{\partial}{\partial x}\left[\mu\left(\frac{\partial u}{\partial z} + \frac{\partial w}{\partial x}\right)\right] + \frac{\partial}{\partial y}\left[\mu\left(\frac{\partial v}{\partial z} + \frac{\partial w}{\partial y}\right)\right]
\end{aligned}
\tag{4}
$$

where:
$\rho$—Air density;
$u$, $v$, $w$—Velocity of air in x, y, z directions;
$f_x$, $f_y$, $f_z$—External body force; and
$\mu$—Dynamic viscosity.

Sun, Chen and Cheng presented Equation (5) to calculate the pressure distribution between two parallel plates in 2005 [13]; it is based on the Navier–Stokes equation, continuity equation and ideal gas law. Since these mathematical models are not able to be solved accurately in most cases, researchers have developed many approximate methods of which one-dimensional analytic method is usually used. Figure 5 illustrates the simplified model of one-dimensional air flow in an aerostatic bearing slideway.

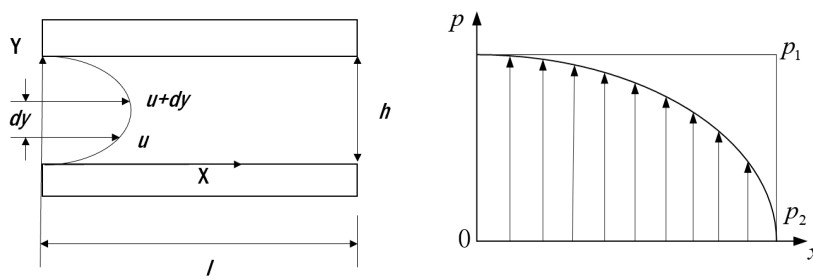

**Figure 5.** One-dimensional model of the parallel plates.

According to the study of Sun et al., based on the one-dimensional assumption, Reynolds equation can be simplified as:

$$\frac{\partial p}{\partial x} = u \frac{\partial^2 u}{\partial y^2}$$
$$\frac{\partial p}{\partial y} = 0 \tag{5}$$
$$\frac{\partial p}{\partial z} = 0$$

Supposing the clearance between two parallel plates is $h$, and then the boundary conditions are:

$$y = 0, \ u = 0;$$
$$y = h, \ u = 0. \tag{6}$$

Insert the ideal gas law equation:

$$\rho = \frac{p}{RT} \tag{7}$$

where:
$T$—the temperature;
$R$—the gas constant;

Omit the mathematical deviation process, the final formula of the pressure distribution is:

$$p_x^2 = p_1^2 - \left(p_1^2 - p_2^2\right)\frac{x}{l} \tag{8}$$

As a short conclusion of this section, mathematical and geometrical models have been established at three different scales, which aim to discuss the influence parameters and factors of the designed aerostatic bearing slideway on the static performance at every single scale. The establishment of multi-scale models can clearly show the design concerns of aerostatic bearing slideway product at various scales and provide a theoretical foundation for subsequent simulations.

## 4. Simulations Development: Results, Analysis and Discussion

### 4.1. Simulation on Static Performance

As a powerful and readily available cross-platform finite element analysis, solver, and multiphysics simulation software, COMSOL is able to simulate, calculate, and analyze the flow of media in an arbitrary geometry. In this study, the CFD module of COMSOL is used to simulate the pressure distribution and load capacity of the air film surface. With the appropriate setup of simulation, the static performance of aerostatic bearing slideway is predictable in advance. The calculation of the CFD module of COMSOL is based on the equation below (9) & (10), and Figure 6 shows the calculation process with a flow chart:
Conservation of momentum:

$$\rho(\mathbf{u} \cdot \nabla)\mathbf{u} = \underbrace{\nabla \cdot \left[-p\mathbf{I} + \mathbf{K}\right]}_{\text{Internal and External pressure}} + \overset{\text{External Force}}{\vec{\mathbf{F}}} \tag{9}$$

Conservation of mass:

$$\rho \nabla \cdot \mathbf{u} = 0 \tag{10}$$

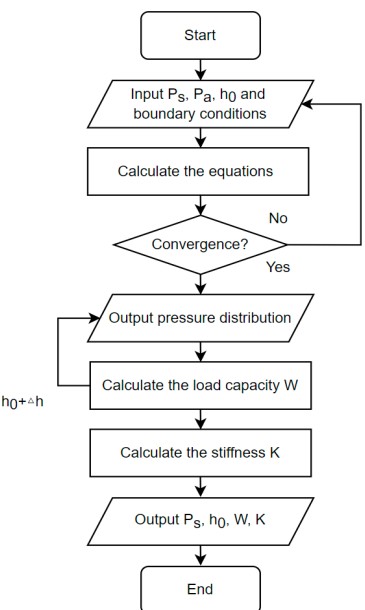

**Figure 6.** Calculation process of COMSOL CFD model.

After establishing the simulation system, the result of a trial run is shown in Figure 7, details of system input are:

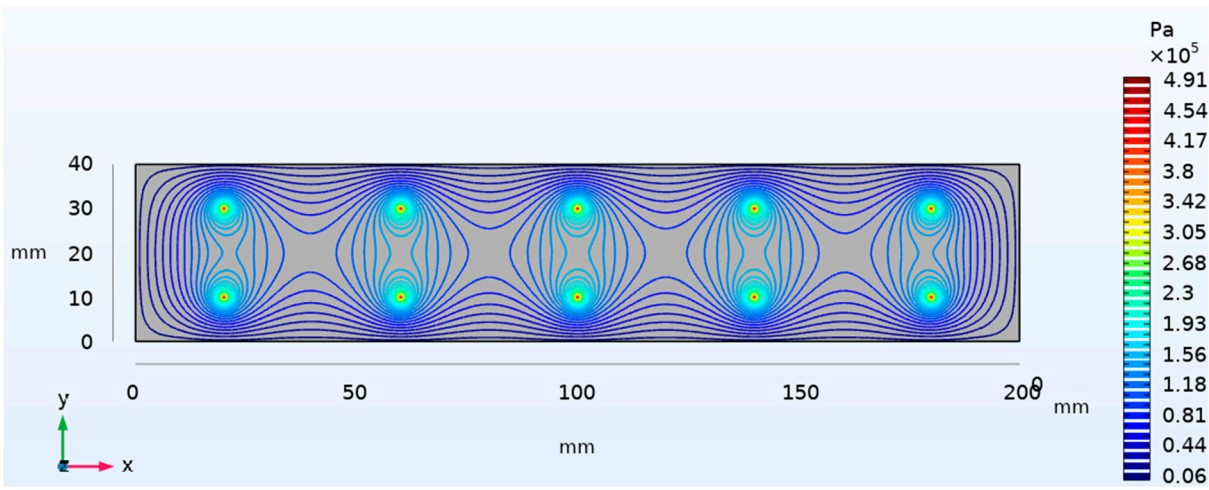

**Figure 7.** Pressure distribution of trial run.

Surface: 40 mm × 200 mm
Supply pressure: 0.5 MPa
Air film thickness: 8 μm
Load-carrying: 491.21 N
Stiffness: 47.95 N/μm

It can be seen that the pressure distribution diagram and related values are able to meet the expectation.

Then running the simulations with different input parameters multiple times, the results are obtained and shown in Figures 8–10 respectively.

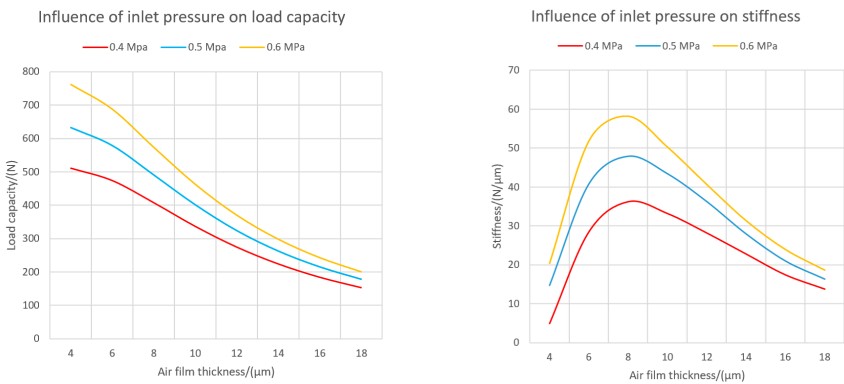

**Figure 8.** Load capacity and stiffness under different supply pressures.

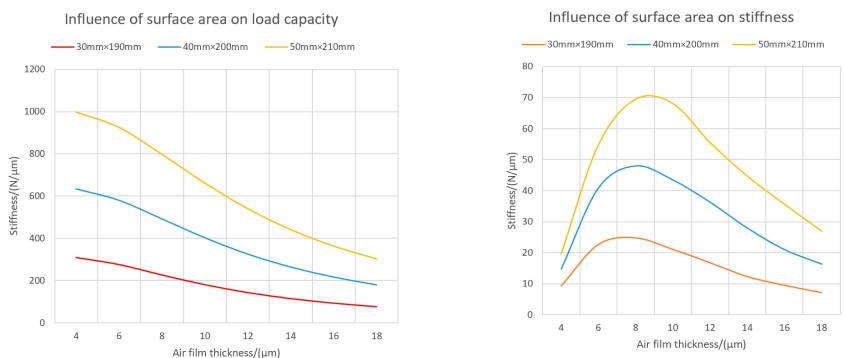

**Figure 9.** Load capacity and stiffness under different air film sizes.

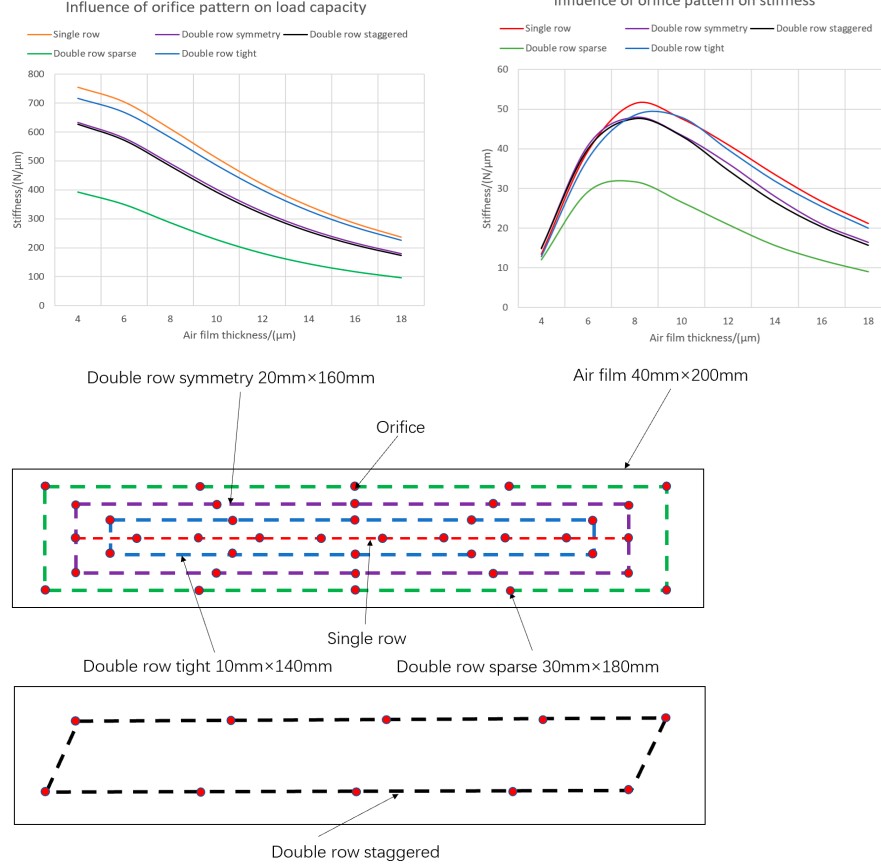

**Figure 10.** Load capacity and stiffness under different orifice patterns.

From these results presented in Figures 8–10, it can be found:

- From the load capacity and stiffness point of view, the bigger air film surface area and higher inlet pressure can bring the slideway better static performance.
- Figures 8 and 9 illustrate that the influence of air film thickness on the slideway's static performance is not linear. No matter what the inlet pressure and surface area were, with the increase of air film thickness, the air film load capacity kept decreasing. However, their stiffness curve can be seen behaving as a parabolic form and reaching the maximum value for the optimum air film thickness of 7–9 μm while depending on the bearing surfaces.
- Under a certain inlet air pressure, the double-row orifice structure showed no benefit on the static performance of the slideway in comparison with single-row orifice. However, according to Tao's research study [14], the high-pressure zone between these two rows of orifices can increase the stability of the slideway during working operations.
- Different orifice patterns can affect the pressure distribution and in future influence the load capacity and stiffness of the air film; the certainty of this law needs more experimental data.

### 4.2. Simulation on Dynamic Performance

After the static performance simulation, the basic design of the aerostatic bearing slideway including structure, parameters, and specifications have been completed and integrated into the virtual model, which plays the role as a fundament of the digital twin. In order to obtain the real-time dynamic performance of the slideway during working, the control system as shown in Figure 11 was constructed:

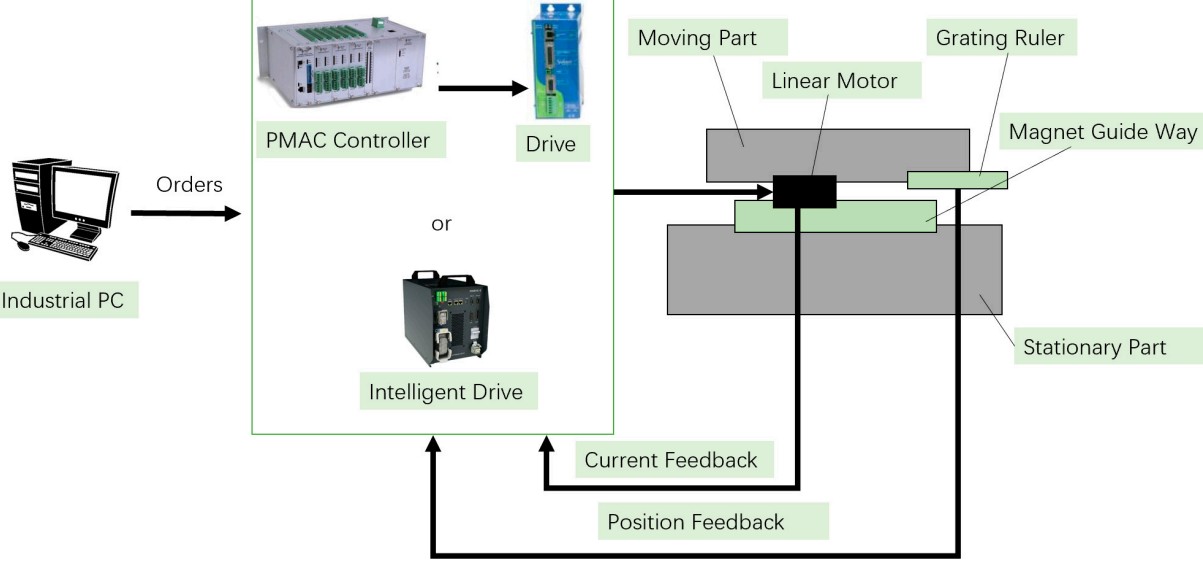

**Figure 11.** Structure of aerostatic bearing slideway control system.

Simulink is a MATLAB-based graphical programming environment for modelling, simulating, and analyzing multidomain dynamical systems, which is widely used in automatic control and digital signal processing for multidomain simulation and model-based design. In this study, the simulation of the dynamic performance e.g., motion accuracy and stability of the aerostatic bearing slideway, is based on the established SIMULINK model. The related linear motor parameter comes from one of the authors—Dr. Dehong Huo; the system details are shown in Figure 12:

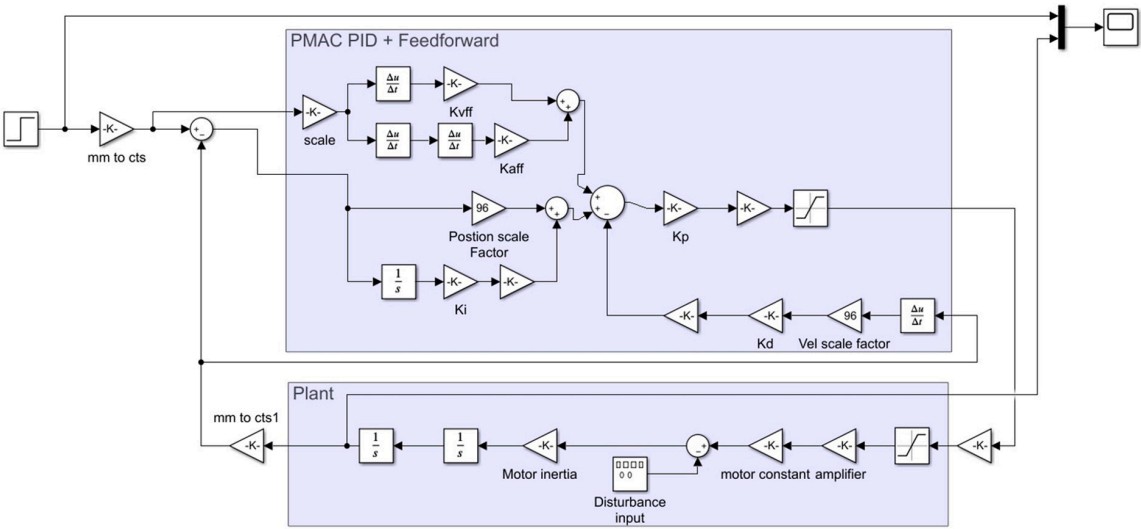

**Figure 12.** SIMULINK system model and subsystem details.

In order to obtain precise control, an open system with PC and PMAC controller is chosen. Due to the study presented in [15], this kind of control system has high scalability and is convenient for integrating subsequent digital twin control systems of other machine tool components. The performance in the aerostatic bearing slideway control system is mainly manifested as motion accuracy and stability, which means the response time, overshoot, and oscillation need to be kept at a low magnitude. In order to perform suitable control of the linear motor, PID controller is thus implemented. After debugging the system with appropriate PID gains, the corresponding response curve can be obtained from the oscilloscope, as shown in Figure 13: The response time of the system to the step signal after debugging is 0.03–0.04 s; the system is relatively stable and has no obvious overshoot, which can meet the demand for ultra-precision motion of the aerostatic bearing slideway; and it has also relatively good stability under the simulated external interference force (50 N, 50 Hz), as shown in Figure 14. Meanwhile, except for the interference force like cutting force or air resistance, aerostatic bearing slideway often bears additional load during practical working. The study on steady state response of the control system under variable loads can help the digital twin system to predict the influence of different load states on the stability and dynamic performance of the slideway system.

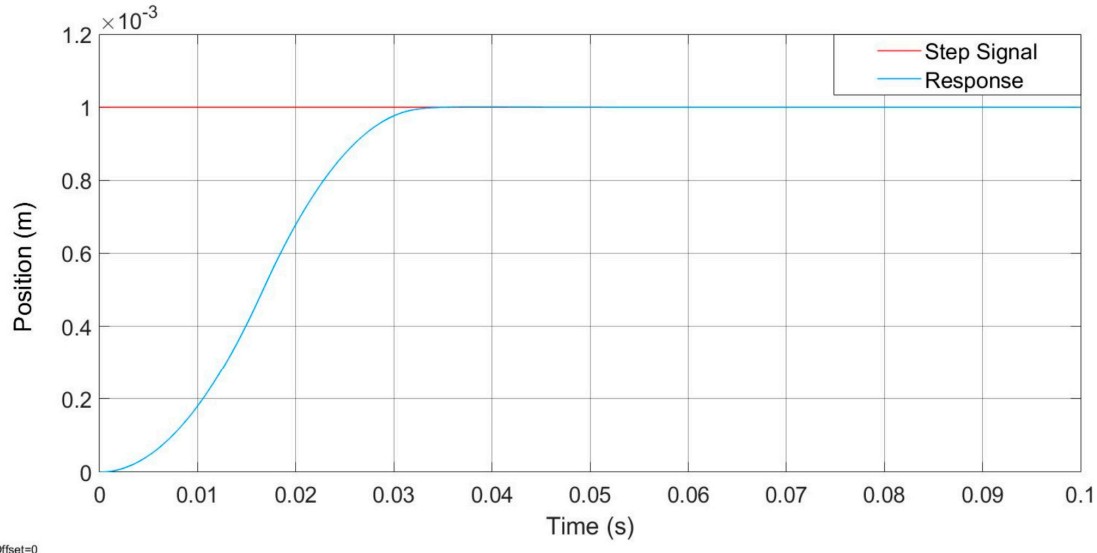

**Figure 13.** System response time to the step signal.

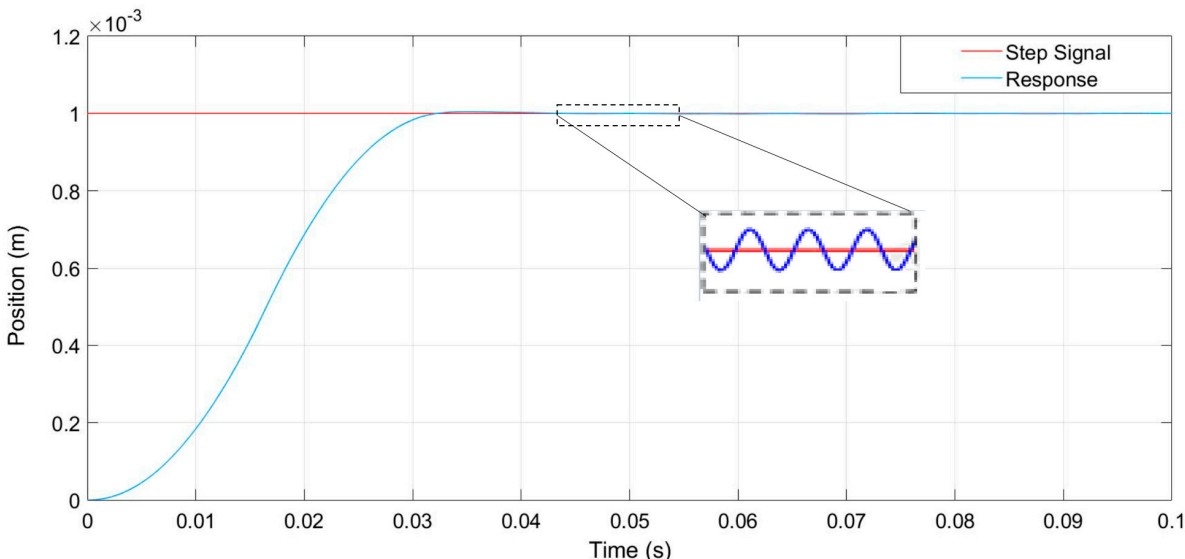

**Figure 14.** System response under external interference.

Figure 15 shows the response time and steady state of the aerostatic bearing slideway control system under 0 kg, 5 kg, 10 kg, and 15 kg loads. It can be found that with the increase of implemented load, the overshoot and response time of the step signal get increased as well. If the load mass is over range, it will cause notable instability of the system and seriously affect the processing quality during manufacturing. Therefore, it is necessary to tune the PID parameters according to different working environments. Since not all end-users have competent control engineering skills and the pre-set automatic tuning algorithm is not applicable to provide the ideal solution against all application scenarios, the remote tuning of the slideway system is thus of significance for both the end-users and the slideway product provider.

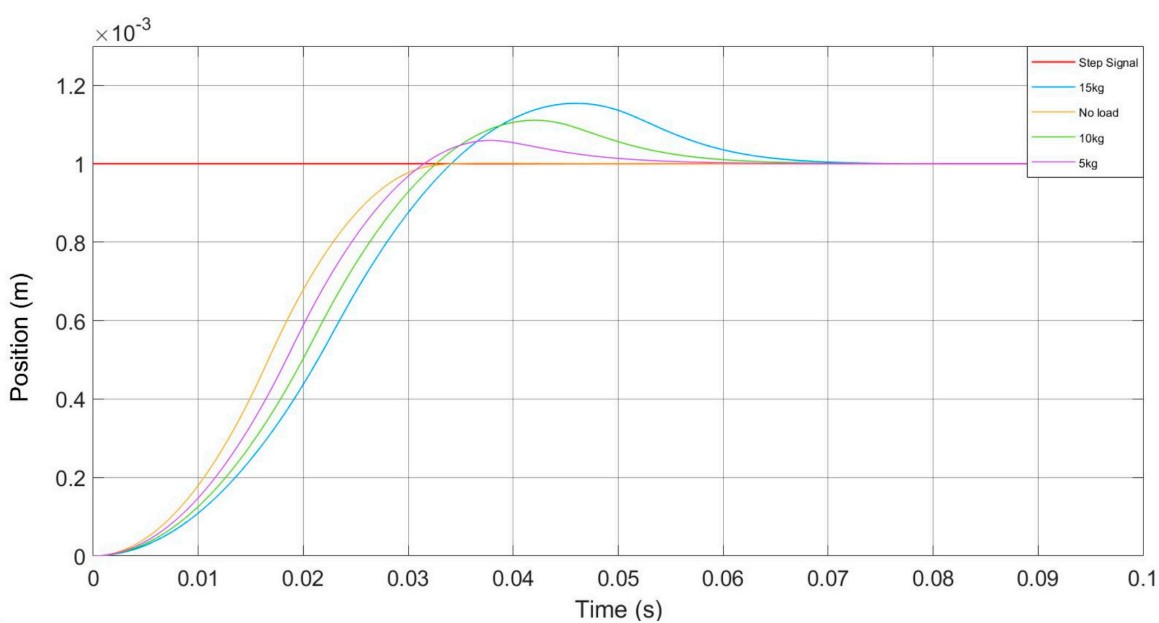

**Figure 15.** The system response time under different loads.

After finishing the design and simulation process of the dynamic performance of aerostatic bearing slideway, the function of remote monitoring and tuning for the digital twin can be further proceeded. By analyzing the real-time data which is fed back, collected

and stored in-situ, as shown in Table 1, the dynamic characteristics in every instantaneous aerostatic bearing slideway can be obtained, and when the system is disturbed by external force or load, the R&D or debugging engineer can remotely adjust the corresponding parameters of the system in order to improve the system stability and dynamic performance of aerostatic bearing slideway, and reduce or solve the performance loss as well, so as to achieve the purpose of continuous improvement of the slideway's dynamic performance.

**Table 1.** Data output from the MATLAB programs.

| Variable 1 | | Variable 2 | |
|---|---|---|---|
| Time (s) | Velocity (m/s) | Time (s) | Position (m) |
| 0 | 0 | 0 | 0 |
| 0.0025 | 0.0092 | 0.0026 | $1.2201 \times 10^{-5}$ |
| 0.0053 | 0.0189 | 0.0054 | $5.1920 \times 10^{-5}$ |
| 0.008 | 0.0287 | 0.0081 | $1.1921 \times 10^{-4}$ |
| 0.0107 | 0.0385 | 0.0109 | $2.1460 \times 10^{-4}$ |
| 0.0134 | 0.0482 | 0.0137 | $3.3646 \times 10^{-4}$ |
| 0.0161 | 0.0579 | 0.0164 | $4.8626 \times 10^{-4}$ |
| 0.0188 | 0.0521 | 0.0192 | $6.3931 \times 10^{-4}$ |
| 0.0215 | 0.0423 | 0.022 | $7.6572 \times 10^{-4}$ |
| 0.0242 | 0.0325 | 0.0247 | $8.6457 \times 10^{-4}$ |
| 0.0269 | 0.0228 | 0.0275 | $9.3587 \times 10^{-4}$ |
| 0.0297 | 0.013 | 0.0303 | $9.7960 \times 10^{-4}$ |
| 0.0324 | 0.0039 | 0.033 | $9.9681 \times 10^{-4}$ |
| 0.0351 | $7.6856 \times 10^{-4}$ | 0.0358 | 0.001 |
| 0.0378 | $6.1492 \times 10^{-5}$ | 0.0386 | 0.001 |

## 5. Aerostatic Bearing Slideway Digital Twin and Its Implementation

The most accepted definition of digital twin is the collection of all digital artifacts that accumulate during product development, integrating all data that is generated during product design and use [16,17].

Figure 16 illustrates the steps and functions of aerostatic bearing slideway digital twin, which can be roughly divided into four steps:

- Step 1: design stage.
  In the design stage of aerostatic bearing slideway, the implementation of digital twins can improve the accuracy of the design and verify the performance of the product in the real environment. The digital twin at this stage only includes the use of CAD tools to develop virtual prototypes of products that meet technical specifications, which accurately record the physical parameters of the product.
- Step 2: simulation stage.
  Through the simulation experiments with repeatable and variable parameters, the static and dynamic performance of aerostatic bearing slideway under different external environments and the adaptability are verified.
- Step 3: manufacturing stage.
  In the manufacturing stage, the use of digital twin can speed up product introduction time, improve product design quality, and reduce expense and time costs.
- Step 4: service stage.
  With the maturity of the Internet of Things technology and the decline in sensor costs, increasing the number of industrial products results in the use of sensors to collect the environment data and working status of the product during the operation. The digital twin at this stage can realize remote monitoring and predictive maintenance. In this study, these functions are realized by feeding back, collecting, and storing real-time data from the grating ruler installed on the moving part of aerostatic bearing slideway, which can help end users avoid malfunctions caused by product misuse, and improve the accuracy of product parameter configuration.

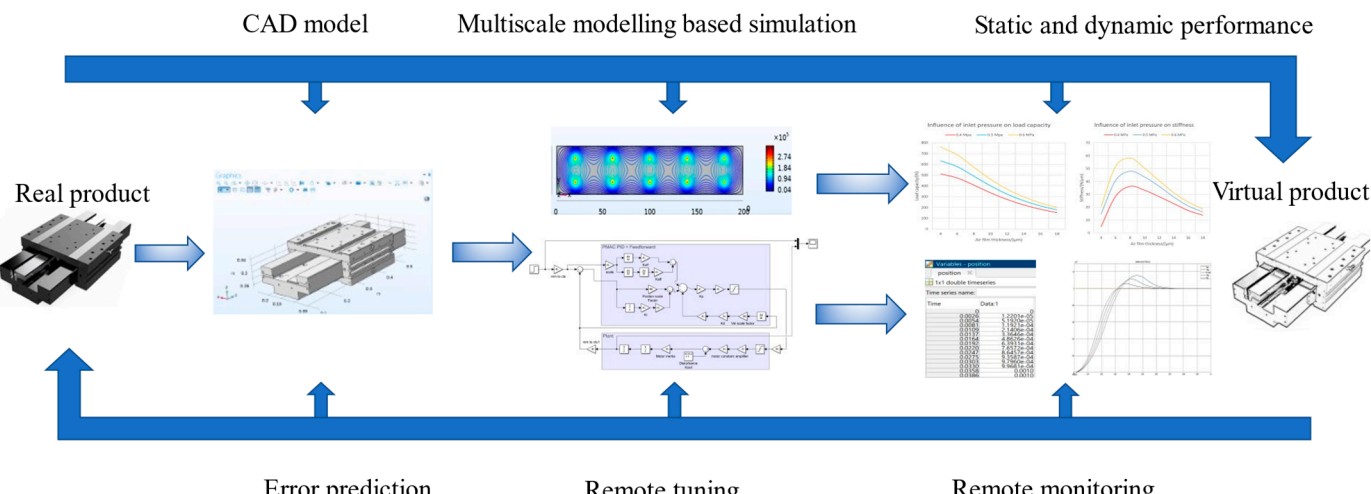

**Figure 16.** Schematic of the aerostatic bearing slideway digital twin.

The application of digital twin technology can provide many practical solutions for aerostatic bearing slideway from implementation perspectives like improving the processing quality of ultra-precision machine and optimizing end-user experience. For instance, the algorithm preset in digital twin system can solve the problem of the decline in machining accuracy caused by the acceleration change of the slideway at the end of the running track; by recording and feeding back the usage data and slideway operating parameters, overload and excessive vibration can be interpreted as early warnings; thus, engineers are able to make corresponding optimization through remote tuning or adjustment of control parameters; the recording data of the full life cycle of aerostatic bearing slideway can help R&D engineers to carry out product iteration and design optimization [18,19].

## 6. Conclusions

The development and results presented in this paper reveal that multiscale modelling and analysis can be used as an enabling methodology for design and development of high-precision engineering products such as aerostatic bearing slideways. Through this approach, it is likely to reveal the intrinsic complex relationships among key design factors working in macro-meso-micro-nano scales and their collective effects on the performance of the slideway system, and to develop a forward-looking design and analysis method for future air-bearing slideway and spindle systems.

The paper also presents the development of the multiscale modelling and analysis approach by using the multiphysics programming environment combined with MATALB simulations. Explorations into the implementation perspectives of this approach have been further discussed, and as an enabling kernel for its extension into digital twin development for the slideway system. For future research work, the approach, development, and underlying fundamentals are expected to be tailorable organically to two-axis and three-axis slideway systems, although this should be thoroughly investigated and expected to report separately in the future.

**Author Contributions:** Conceptualization, K.C. and N.G.; methodology, K.C., D.H. and N.G.; software, N.G.; writing—original draft preparation, N.G.; writing—review and editing, K.C.; supervision, K.C. and D.H. All authors have read and agreed to the published version of the manuscript.

**Funding:** This research received no external funding.

**Data Availability Statement:** The data presented in this study are available on request from the corresponding author.

**Conflicts of Interest:** The authors declare no conflict of interest.

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
