# Peer review of "Multiscale Modelling and Analysis for Design and Development of a High-Precision Aerostatic Bearing Slideway and Its Digital Twin"

_machines, doi:10.3390/machines9050085_

Round 1
Reviewer 1 Report
Presented work is valuable and can be interested in others but have minor shortcomings. Authors describe multiscale modeling aerostatic bearing. This topic is timely. However, the conclusions should be an improvement. Authors should give more recommendations for designers how would like to design such components. Also, paragraph 4.1 should be more explained in conclusion 3. Maybe simulation with the non-evenly load.
Particular comments are the following:
Variables are written in a simple font and in other places in italics style. Please correct it.
Line 155-157 Number of the equation was moved into text.
Line 264 Pressure should be in MPa.
Figure 13, 14 & 15 Larger font of the legend should be. Additional titles of the axis should be added.
Author Response
We are very grateful for your valuable comments and thoughtful handling of the manuscript, which have undoubtedly helped to improve the substance and readability of our paper. All of your comments and suggestions have been addressed, and the manuscript has been revised substantially. We appreciate the opportunity to make use of these constructive comments to strengthen the paper. In the attached response, we have addressed each of your comments on a point-by-point basis, and all of the revised parts have been highlighted in red.

Reviewer 2 Report
The paper optimized an aerostatic bearing slideway by using multiscale modeling and analysis. MATLAB, Simulink, and COMSOL (Computational Fluid Dynamics - CFD) simulations revealed the influence of the main parameters (air film thickness and orifice pattern) on the load capacity and stiffness of the aerostatic bearing slideway.
The paper is original and the research is well conducted, but it can be further improved before being recommended for publication. My suggestions and questions are:
- It can be observed from Figures 8 and 9 that there is an optimum of air film thickness from the viewpoint of the stiffness, no matter the inlet pressure and the surface area. The air film thickness, in my opinion, results from the equilibrium of applied load and aerostatic pressure force. The simulations are truncated. This aspect should be argued and discussed.
- Remarks 1 and 2, presented on page 9, lines 286-289 are wrong. Remark 1 should emphasize that there is an optimum air film thickness. The assertion that smaller film thickness assures better load capacity and stiffness is wrong if we observe the results from Figures 8, 9, and 10 (the right side). From the viewpoint of load capacity and stiffness, this remark should be developed more, for each simulation apart. For remark 2, not the air film thickness reached a peak, but the stiffness reached maximum values for the optimum air film thickness of 8-9 microns.
- Increase the legend text and the text of the titles of figures 13 and 14.
- On page 12, lines 352-355, why there is the need to manually tune or adjust the control parameters of the system when an automatic system could do this?
- The measuring unit of the ordinate axis in Figure 7 is missing.
Author Response

(The authors gave the same response as above.)

Reviewer 3 Report
The subject of the article is high precision aerostatic bearing slideway. These types of bearing slideways are very interesting. The introduction is imprecise, in particular when it comes to the formulation of the problem that is analyzed and the contribution of the authors to this solution.
Multiscale modeling applied to the aerostatic bearing has been insufficiently described. Much of the information is of a very general nature, insufficient for a good understanding of the content of the article.
There are quite a few illustrative pictures and charts that have been well covered. Some pictures are difficult to read (Fig. 12).
The conclusions presented in the article are supported by the content of the article, but at the same time reflect its shortcomings. The analyzed problem is fuzzy, the authors' contribution to solving this problem is not entirely clear, and the information provided in the content is incomplete, i.e. for example, it does not allow the reader to reproduce the results on his own.
I believe the work can be a bit polished.
Author Response

(The authors gave the same response as above.)
